# Fetal malposition in labour and health outcomes for women and their newborn infants: A retrospective cohort study

Jennifer Barrowclough[1,2]*, Bridget Kool[3], Caroline Crowther[1]

1 Liggins Institute, University of Auckland, Auckland, New Zealand, 2 Department of Midwifery, School of Clinical Sciences, Auckland University of Technology, Auckland, New Zealand, 3 School of Population Health, Faculty of Medical and Health Sciences, University of Auckland, Auckland, New Zealand

* j.barrowclough@auckland.ac.nz

**Data Availability Statement:** All relevant data are within the paper.

**Funding:** J.B received a PhD scholarship from Liggins Institute, University of Auckland, New

## Abstract

### Introduction

Occiput-posterior (OP) or occiput-transverse (OT) fetal malposition has a prevalence of 33–58% in the first-stage of labour with 12–22% persisting until delivery. Malposition is associated with significant maternal and neonatal morbidity. Most previous studies report the incidence and adverse maternal and fetal outcomes of persistent fetal malposition in the second stage of labour and do not include outcomes that may be present in the first stage of labour.

### Aims

To assess the incidence and health outcomes for women and their newborn infants of a fetal malposition in the first or second stage of labour.

### Materials and methods

A retrospective cohort study of 738 maternity records (randomly selected) from a tertiary hospital in New Zealand. Maternal and neonatal characteristics are described. Outcomes for women with a fetus in an OP or OT position in labour are compared to those for women with a fetus in an occiput-anterior position (OA).

### Results

499 (68%) women had an OP/OT positioned fetus and 239 (32%) had an OA positioned fetus on vaginal examination in labour. Women had similar characteristics except a body mass index $\geq$30 kg/m² was more common in the OP/OT group. Fetal malposition appears to be more likely in women with a right-sided fetal occiput. Three quarters of OP/OT fetuses rotated anteriorly by birth. Fetal malposition compared to no malposition was associated with oxytocin augmentation, epidural use, a longer first stage of labour, fewer normal vaginal births, and more caesarean sections. Fetal malposition during labour was not associated with adverse neonatal outcomes.

Zealand and Shundi Group Ltd, and payment of University fees by Work Force New Zealand. The funders had no role in the study design, data collection and analysis, decision to publish, or preparation of the manuscript.

**Competing interests:** The authors have declared that no competing interests exist.

## Conclusion

Interventions such as maternal posture in the first and second stage of labour could potentially reduce the incidence of malposition and improve health outcomes for mothers.

## Introduction

Fetal malposition refers to a fetus in an occiput-posterior (OP) or occiput-transverse (OT) position in labour [1]. In the first stage of labour fetal malposition has a prevalence of between 33–58%, with 12–22% remaining as a persistent malposition at delivery [2, 3]. Right-sided fetal malposition is approximately twice as prevalent as left-sided malposition [4, 5], considered due to dextrorotation of the uterus and location of the sigmoid colon on the maternal left [6, 7]. Factors associated with fetal malposition include nulliparity [8], an anterior placenta [9], pelvic shape [10], epidural use [11], increased body mass index [2], advanced maternal age and fetal macrosomia [12]. Restricted space for anterior fetal rotation may occur due to strong abdominal muscles in nulliparous women or fetal macrosomia. Alternatively, slower progression to full cervical dilatation and therefore exposure of the fetal head to counterpressure of the pelvic floor is another possible mechanism by which nulliparity is associated with persistent malposition.

Persistent fetal malposition is associated with adverse maternal health outcomes including operative vaginal birth, caesarean section [2, 8, 13–16], postpartum haemorrhage, endometritis, chorioamnionitis [12], severe perineal injury [8, 14, 16, 17] and anal sphincter injury [8]. For the neonate, malposition is associated with admission to a neonatal intensive care unit (NICU), birth injury [18] including sub-galeal haematoma [19], and hypoxic ischaemic encephalopathy [20].

Studies assessing fetal malposition commonly report outcomes of persistent fetal malposition in the second stage of labour [14–16], and therefore do not include women in labour who do not progress to the second stage who may experience a range of adverse outcomes. Although two studies that assessed malposition in early labour reported on caesarean section [3, 21], and duration of first stage labour [3], the evidence dates 20 years and practices may have changed. Other studies reporting outcomes of fetal malposition in early labour assessed fetal position at induction of labour [13, 22, 23], which was not associated with adverse labour and birth outcomes [22, 23]. Therefore, the aim of this study was to assess the incidence and health outcomes for women and their newborn infants with a fetal malposition in the first or second stage of labour.

## Method and materials

A retrospective cohort of women who laboured and gave birth at Auckland City Hospital, a tertiary hospital in New Zealand, in 2018. Women ≥ 16 years of age with a singleton, cephalic presentation who were induced into or were in established labour at term (≥37 weeks gestation) and for whom fetal position in first and/or second stage of labour had been determined by vaginal examination by a midwife or obstetrician were eligible for inclusion. Fetal position was only extracted for women with a cervical dilatation ≥3cm and regular contractions. Women with a major fetal abnormality were not included.

The following information was sought from routinely collected and hand searched electronic data: maternal characteristics (age, ethnicity, area deprivation scale [NZDep 2013] [24],

body mass index [BMI] at initial prenatal clinic, parity, previous caesarean, gestational age, lead maternity carer (midwife or obstetric specialist contracted to provide maternity care) during pregnancy, and type of midwife assisting in labour); labour characteristics (anterior placenta, pre-labour rupture of membranes [ROM], artificial ROM, duration of ROM, liquor colour and volume in labour, induction of labour, oxytocin augmentation, epidural analgesia, labour duration, intrapartum pyrexia, left/right-sided fetal occipital location, labour remedies (Rebozo–jiggling the woman's suspended abdomen with fabric, acupuncture, reflexology, massage, hypnobirthing, homeopathy, Swiss ball, peanut-ball, stretches), use of manual rotation, fetal position in first and/or second stage labour and at birth); birth characteristics (mode of birth, type of operative vaginal birth, caesarean indication, perineal trauma, shoulder dystocia, blood loss, postpartum haemorrhage [PPH], any postpartum urinary catheterisation (as a measure of maternal complications), urinary tract infection [UTI], wound infection (episiotomy or abdominal), high dependency/critical care [HDU/CCU] admission, duration of postnatal hospital stay); neonatal characteristics (stillbirth, neonatal death, resuscitation required, Apgar score at <7 at 5 minutes, umbilical arterial lactate $\geq$ 6 mmol/l, birth weight, growth centile >90th, head circumference, length, NICU admission, respiratory distress syndrome [RDS] [25], non-specific respiratory distress [25], hypoglycaemia <2.6 mmol/l prior to hospital discharge, jaundice requiring phototherapy, duration of NICU stay, and feeding method at hospital discharge). Abnormal umbilical arterial lactates, considered a more accurate measure of metabolic acidosis than umbilical artery pH <7.0 and Base Excess <-12 mmol/l alone, were the standard measure of fetal acidosis at the hospital in 2018 with a clinical cutoff of $\geq$ 6 mmol/l, though revised to $\geq$6.1 mmol/l in 2019 [26]. The NZDep is an area-based measure of socioeconomic deprivation in New Zealand based on deciles, Decile 1 represents areas with the least deprived scores, Decile 10 represents areas with the most deprived scores. Data concerning fetal position at caesarean birth was collected from the operation notes.

The primary study outcome was caesarean section. A power calculation performed before the start of the study using ClincCalc.com determined a sample size of 1000 would detect an incidence of caesarean section of 16% in the OP/OT group and 10% in the occiput-anterior (OA) group [21] with 80% power and 95% confidence level. A total of 1000 women were selected using random sampling, conducted by an independent statistician using R statistical packages (R Core Team 2013), from the 4376 eligible women who gave birth at Auckland Hospital during 2018. However, 262 maternity records had no record of fetal position in labour so were unable to provide data, resulting in a sample of 738 women. Secondary outcomes included a range of labour, birth and neonatal characteristics.

Comparisons of continuous variables for OP/OT and OA groups were performed using the two-tailed Student's t-test (SPSS for Windows version 27, SPSS Inc., Armonk, NY, USA). Comparisons of categorical variables were performed using chi squared and Fisher exact tests where appropriate (Epi Info v7.2.4), and SPSS for subset variables. Values were expressed as number, percentage, risk ratio (RR) with 95% confidence intervals (CI) and *P*-values, mean (m) ± standard deviation (SD) and mean difference (MD). A P-value ≤0.05 was used to denote a statistically significant difference. Confounding of the outcome artificial rupture of membranes (ARM) was controlled for the use of oxytocin augmentation using a general linear model for the univariate analysis with OP/OT position as the fixed factor, ARM as the dependant variable, and oxytocin augmentation as the confounding co-variate.

Ethical approval for the study was obtained from the Auckland Health Research Ethics Committee (Reference: 000133), and the 'Research Governance Group for Women's Health and Neonatal' provided approval. Data were deidentified in Excel once linkage of electronic data to hand-searched data was complete. Ethics approval was provided for the use of data from women aged 16 years+ without consent because the data sharing is covered by Auckland

District Health Board's data access policy. The data was de-identified during analysis and reporting, and is only used for research purposes intended to benefit maternal and neonatal health in accordance with the Declaration of Helsinki 1996.

## Results

In 499 (68%) cases the baby was in an OP/OT position and in 239 (32%) in an OA position either during the first or second stage of labour (Table 1).

### Maternal characteristics

The mean maternal age for both the OP/OT and OA groups was 31 years ±SD 5 years and the gestational age at birth was similar (39.3 and 39.2 weeks ±SD 1.1 for OP/OT and OA respectively). There were no differences between the two groups by ethnicity, deprivation, parity, type of midwifery care in labour, type of lead maternity carer, or history of a previous caesarean (Table 1). There were differences between the two groups by BMI ($P = 0.013$). Women in the OP/OT group were more likely to have a BMI $\geq 30$ kg/m$^2$ (RR 1.56, 95% CI 1.04–2.34), and less likely to have a BMI $< 25$ kg/m$^2$ (RR 0.85, 95% CI 0.76–0.95) compared to women in the OA group.

### Labour characteristics

Women in the OP/OT group were more likely to receive oxytocin augmentation in labour (51% cf. 38%, $P<0.001$), and have an epidural for analgesia (74% cf. 61%, $P<0.001$) than women in the OA group (Table 2). The first stage of labour was longer for women in the OP/OT group (MD 2:32 hours, $P = 0.001$). There were differences between the groups by fetal occipital location during labour ($P = <0.001$), with women in the OP/OT group more likely to have a baby in a right occipital location (39% cf. 14%), and less likely to have their baby in a left occipital location (35% cf. 44%) or have a direct or undetermined location (26% cf. 41%) than women in the OA group.

There was no difference between the OP/OT and OA groups for the following variables: anterior placenta, pre-labour rupture of membranes (ROM), artificial ROM, duration of ROM, liquor colour, liquor volume, IOL, second stage labour duration, pyrexia in labour, or use of labour care remedies. After controlling for confounding by oxytocin augmentation, artificial ROM was associated with fetal malposition (P = 0.050).

Manual rotation to correct fetal malposition during the second stage of labour was successful in over half (n = 16/30, 53%) of attempts in the OP/OT group and was not attempted in the OA group.

### Birth characteristics

There were differences between the groups by fetal position at birth (P<0.001) (Table 3). For the 499 (67%) women with an OP/OT fetal position, 74% rotated anteriorly by the time of birth and 26% remained in a malposition (9% OT and 17% OP). Of the 239 (32%) women in the OA group, 94% had a fetus remain in the OA position and only 4% had a fetus rotate to an OP/OT position by the time of birth. Overall, 18% of fetuses (136/738) from both the OP/OT group (n = 127) and OA group (n = 9) were in an OP/OT malposition at birth. Women in the OP/OT group compared to women in the OA group were more likely to give birth by caesarean section (RR 3.0, 95% CI 1.90–4.75, $P<0.001$), and less likely to have a spontaneous vaginal birth (RR 0.85, 95% CI 0.74–0.97, $P = 0.017$), or have an episiotomy (RR 0.80, 95% CI 0.65–0.99, $P = 0.045$). Overall, there was no difference in the need for an operative vaginal birth (RR

**Table 1. Maternal characteristics for women in labour with an occiput-posterior (OP) or occiput-transverse (OT) positioned fetus compared with an occiput-anterior (OA) positioned fetus.**

| Characteristic<br>Total n(%) OP/OT; OA† | OP/OT in labour<br>n = 499 (67.6%)<br>n(%) | OA in labour<br>n = 239 (32.4%)<br>n(%) | P-value |
|---|---|---|---|
| **Maternal age:** | | | 0.231 |
| <20 years | 11 (2.2) | 7 (2.9) | |
| 20–30 years | 193 (38.7) | 106 (44.4) | |
| 31–40 years | 280 (56.1) | 123 (51.5) | |
| ≥ 41 years | 15 (3.0) | 3 (1.3) | |
| **Ethnicity‡:** | | | 0.289 |
| Māori | 26 (5.2) | 12 (5.0) | |
| Pacific Peoples | 61 (12.2) | 22 (9.2) | |
| European | 205 (41.1) | 94 (39.3) | |
| Asian | 173 (34.7) | 100 (41.8) | |
| Middle Eastern/Latin American/African | 34 (6.8) | 11 (4.6) | |
| **Area deprivation scale§:**<br>499 (100); 237 (99.2) | | | 0.245 |
| Score 1–2 | 79 (15.8) | 37 (15.5) | |
| Score 3–4 | 91 (18.2) | 52 (21.8) | |
| Score 5–6 | 125 (25.1) | 50 (20.9) | |
| Score 7–8 | 92 (18.4) | 55 (23.0) | |
| Score 9–10 | 112 (22.4) | 43 (17.9) | |
| **Body Mass Index (BMI):**<br>496 (99.4); 237 (99.2) | | | 0.013 |
| BMI < 25 | 300 (60.1) | 169 (71.3) | |
| BMI 25–29 | 108 (21.6) | 41 (17.3) | |
| BMI ≥ 30 | 88 (17.6) | 27 (11.4) | |
| **Parity:** | | | 0.518 |
| Parity 0 | 298 (59.7) | 150 (62.8) | |
| Parity 1–2 | 184 (36.9) | 84 (35.1) | |
| Parity ≥3 | 17 (3.4) | 5 (2.1) | |
| **Previous caesarean** | 34 (6.8) | 12 (5.0) | 0.346 |
| **Gestational Age:** | | | 0.664 |
| 37 to 38 weeks | 133 (26.7) | 67 (28.0) | |
| 39 to 40 weeks | 298 (59.7) | 145 (60.7) | |
| ≥ 41 weeks | 68 (13.7) | 27 (11.3) | |
| **Type of lead maternity carer in pregnancy:** | | | 0.831 |
| Self-employed Midwife | 229 (45.9) | 114 (47.9) | |
| Hospital Team¶ | 141 (28.3) | 67 (28.2) | |
| Private Obstetrician | 129 (25.9) | 57 (23.9) | |
| **Midwife assisting in labour:** | | | 0.857 |
| Core midwife†† | 206 (41.3) | 97 (40.6) | |
| Self employed | 293 (58.7) | 142 (59.4) | |

† Total OP/OT n (%), OA n (%) for variable where different from whole sample.

‡ Stats NZ Level 1 ethnicity.

§NZDep 2013.

¶Midwifery/obstetric team supervised by a senior medical officer.

††Hospital employee.

**Table 2. Comparison of labour characteristics for women with an occiput-posterior (OP) or occiput-transverse (OT) positioned fetus compared with an occiput-anterior (OA) positioned fetus.**

| Labour Variables<br>Total n(%) OP/OT; OA† | OP/OT in labour<br>499 (67.8)<br>n(%) | OA in labour<br>239 (32.4)<br>n(%) | Risk Ratio (RR)/<br>Mean Difference (MD) (95% CI) | P-<br>Value |
|---|---|---|---|---|
| **Anterior placenta**<br>411 (82); 201 (84) | 219 (53.3) | 95 (47.3) | 1.13<br>(0.95–1.34) | 0.162 |
| **Pre-labour rupture of membranes‡** | 37 (7.4) | 26 (10.9) | 0.69<br>(0.42–1.09) | 0.115 |
| **Artificial rupture of membranes**<br>495 (99); 233 (97) | 319 (64.4) | 133 (57.1) | 1.13<br>(0.99–1.28) | 0.056 |
| **Duration membranes ruptured**<br>**(hour: min) mean ±SD**<br>498 (99.8); 238 (99.6) | 11:34 ±SD 16:37 | 11:38 ± SD 32:44 | MD -0:04<br>(-4:30–4:22) | 0.975 |
| **Liquor colour in labour:**<br>473 (95); 227 (95) | | | | 0.747 |
| Clear/blood stained | 385 (81.4) | 194 (85.5) | | |
| Meconium thin | 31 (6.5) | 13 (5.7) | | |
| Meconium moderate/thick | 57 (12.1) | 20 (8.8) | | |
| **Liquor volume in labour:**<br>395 (79.2); 196 (82.0) | | | | 0.274 |
| Normal | 358 (90.6) | 174 (88.8) | | |
| Absent or reduced | 26 (6.6) | 19 (9.7) | | |
| Excessive | 11 (2.8) | 3 (1.5) | | |
| **Induction of Labour** | 266 (53.3) | 124 (51.9) | 1.03<br>(0.89–1.19) | 0.717 |
| **Oxytocin augmentation** | 253 (50.7) | 90 (37.7) | 1.35<br>(1.12–1.62) | <0.001 |
| **Epidural anaesthesia** | 367 (73.5) | 146 (61.1) | 1.2<br>(1.07–1.35) | <0.001 |
| **Labour first stage (hour: min)**<br>**mean ±SD**<br>406 (81.4); 217 (90.8) | 13:59<br>±SD 11:28 | 11:27 ±SD 8:04 | MD 2:32<br>(0:59–4:05) | 0.001 |
| **Labour second stage (hour: min)**<br>**mean ±SD**<br>358 (71.7); 182 (76.2) | 1:30 ±SD 1:18 | 1:24 ±SD 1:11 | MD 0:06<br>(-0:07–0:19) | 0.377 |
| **Pyrexia in labour** | 19 (3.8) | 6 (2.5) | 1.52<br>(0.61–3.75) | 0.362 |
| **Fetal occiput location:** | | | | <0.001 |
| Left side | 172 (34.5) | 106 (44.4) | | |
| Right side | 195 (39.1) | 34 (14.2) | | |
| Direct or undetermined side | 132 (26.4) | 99 (41.4) | | |
| **Labour remedies:**<br>47 (9.8); 16 (6.7) | | | | 0.740* |
| Alternative remedies§ | 13 (27.7) | 3 (18.8) | | |
| Swiss ball/rocker/peanut-ball/stretches/rebozo | 34 (72.3) | 13 (81.3) | | |

†Total OP/OT n (%), OA n (%) for variable where it differs from whole sample

‡Prelabour rupture of membranes requiring induction of labour.

§Alternative remedies include massage, hypno- birthing, acupuncture, reflexology, homeopathy.

*Fisher Exact P value for small size values.

**Table 3. Birth characteristics for 738 women in labour with an occiput-posterior (OP) or occiput-transverse (OT) positioned fetus compared with an occiput-anterior (OA) positioned fetus.**

| Birth Variables Total n(%) OP/OT; OA† | OP/OT in labour n = 499 (67.6%) n(%) | OA in labour n = 239 (32.4%) n(%) | P-value |
|---|---|---|---|
| **Fetal position at birth:** | | | <0.001* |
| OA | 367 (73.5) | 224 (93.7) | |
| OP/OT | 127 (25.5) | 9 (3.8) | |
| Other cephalic‡ | 5 (1.0) | 6 (2.5) | |
| **Mode of birth:** | | | <0.001 |
| Spontaneous vaginal birth | 256 (51.3) | 145 (60.6) | |
| Operative vaginal birth | 124 (24.8) | 75 (31.3) | |
| Emergency caesarean section | 119 (23.8) | 19 (7.9) | |
| **Type of operative vaginal birth:** 124 (24.8); 75 (31.4) | | | 0.011* |
| Non-rotational instrumental | 110 (88.7) | 74 (98.7) | |
| Rotational instrumental | 14 (11.3) | 1 (1.3) | |
| **Caesarean indication:** 119 (23.8); 19 (7.9) | | | 0.206* |
| Fetal distress | 23 (19.3) | 7 (36.8) | |
| Inefficient uterine action | 25 (21.0) | 2 (10.5) | |
| Obstructed labour | 71 (59.7) | 10 (52.6) | |
| **Perineal trauma:** 420 (84.1); 229 (95.8) | | | 0.123 |
| Perineum intact | 83 (19.8) | 27 (11.8) | |
| 1˚tear or graze | 57 (13.6) | 31 (13.5) | |
| 2˚ tear | 113 (26.9) | 69 (30.1) | |
| 3˚ tear or more | 16 (3.8) | 12 (5.2) | |
| Episiotomy | 151 (30.2) | 90 (37.7) | |
| **Shoulder dystocia** | 20 (4.0) | 7 (2.9) | 0.465 |
| **Blood loss (mls) mean ±SD** | 528 ± SD 447 | 495 ± SD 438 | 0.347 |
| **Any postpartum haemorrhage:** | 118 (23.6) | 66 (27.6) | 0.244 |
| **Postpartum haemorrhage by birth:** | | | 0.048 |
| Postpartum haemorrhage/Caesarean§ | 24 (20.3) | 6 (9.1) | |
| Postpartum haemorrhage/Vaginal birth§ | 94 (79.7) | 60 (90.9) | |
| **Major postpartum haemorrhage (>1500mls)** | 18 (3.6) | 13 (5.4) | 0.245 |
| **Urinary catheter sited postnatally** | 48 (9.6) | 35 (14.6) | 0.043 |
| **Urinary infection** | 9 (1.9) | 9 (3.8) | 0.106 |
| **Maternal postnatal stay (days) mean ±SD** | 1.44 ±SD 1.98 | 1.15 ±SD 1.92 | 0.053 |

†Total OP/OT n (%), OA n (%) for variable where it differs from whole sample.

‡Other cephalic includes brow n = 1, vertex, and hand n = 12, vertex unspecified n = 1 with no documented fetal position.

§PPH ≥ 500mls if vaginal birth; PPH >1000mls if caesarean section.

*Fisher Exact P value used due to small sample size.

0.79, 95% CI 0.62–1.01, $P = 0.061$), although there were more rotational operative births in the OP/OT group compared to the OA group ($P = 0.023$ Fisher Exact Test). There was no difference in the overall rate of postpartum haemorrhage between the fetal position groups, although there were more PPH related to caesarean in the OP/OT group than the OA group (RR 2.24, 95% CI 0.96–5.19, $P = 0.048$). Fewer postnatal catheterisations were performed for women in the OP/OT group (9.6% cf. 14.6% OA, $P = 0.043$).

No differences were seen between the OP/OT and OA groups for the following: caesarean indication, overall perineal trauma, shoulder dystocia, mean blood loss, major PPH, and duration of postnatal stay. Numbers for failed operative vaginal birth, wound infection, and admission to HDU/CCU were too low in either group to assess differences between the groups.

Further analysis revealed caesarean section was similarly more often in the first stage of labour: n = 83 (70%) OP/OT cf n = 14 (74%) OA, P = 0.727; and this did not differ by nulliparity: n = 62 (74.7%) OP/OT cf n = 13 (93%) OA, P = 0.179 (Fisher Exact Test).

### Neonatal characteristics

There were no differences between the OP/OT and OA groups for any of the neonatal outcomes reviewed including: live birth, resuscitation at birth, Apgar score <7 at 5 minutes, umbilical arterial lactate ≥ 6.0 mmol/l, birth weight, head circumference, length, customised growth >90th centile, NICU admission, duration of NICU stay, RDS, non-specific respiratory distress, hypoglycaemia <2.6 mmol/l, jaundice requiring phototherapy, birth injury, or method of neonatal feeding on discharge. (Table 4). There were no neonatal deaths.

## Discussion

The overall incidence of malposition was 68% in labour and 18% at birth. Possible reasons for the greater frequency of OP/OT position compared to OA position during labour include the presence of risk factors associated with fetal malposition, and the presence of malposition at the establishment of induced labour [27], a process that often involves 24–48 hours of inpatient care for fetal monitoring commonly in semi-recumbent postures in which gravity may encourage posterior rotation of the fetal spine.

Key associations with malposition were high maternal BMI and a right fetal occiput position. Women with a fetal malposition were more likely to have their labour augmented with oxytocin, an epidural analgesia, a longer first stage labour, and give birth by rotational operative vaginal delivery or emergency caesarean section compared with women without a fetal malposition. Fetal malposition was associated with fewer episiotomies and postnatal urinary catheterisations. None of the neonatal outcomes assessed were associated with fetal malposition in labour.

A strength of this study is the novel approach of assessing outcomes related to malposition that is diagnosed during first and second stage labour rather than only a persistent OP/OT in the second stage labour or at birth, allowing assessment of outcomes before a spontaneous anterior rotation. The findings provide contemporary information about fetal malposition in labour including maternal demographics, specifics of labour, and maternal and neonatal health outcomes from the first and second stages of labour. Whilst digital diagnosis of fetal position has a reported inaccuracy of 14–41% in the first stage of labour [3, 28] and 23–27% in the second stage [3, 29] compared to sonographic diagnosis, it remains the standard method of assessing fetal position in labour, enabling this study's outcomes to be benchmarked to standard assessment of fetal position in the clinical setting.

The findings should be considered in the light of several limitations. This was a descriptive study and therefore cannot establish causality. The study inherently has a reduced OP/OT sample by the time of birth due to anterior rotations during labour, and so it carries the risk of under-reporting any associated adverse outcomes rather than over-reporting. As previously described, vaginal examination of fetal position is less accurate than sonographic diagnosis. Data were sometimes not available for some of the study outcomes due to an absence of documentation in the medical records. Whilst missing data are not reported as a subset category, the denominators reflect the totals minus the missing data. Though hand-written notes could

**Table 4. Characteristics for neonates born to women who had an occiput-posterior (OP) or occiput-transverse (OT) positioned fetus compared with an occiput-anterior (OA) positioned fetus.**

| Neonatal outcomes<br>Total n(%) OP/OT; OA† | OP/OT in labour n/499 (67.6%)<br>n (%) | OA in labour n/239 (32.4%)<br>n (%) | *P* value |
|---|---|---|---|
| **Live birth yes** | 499 (100) | 238 (99.6) | 0.324* |
| **Resuscitation at birth** | 35 (7.0) | 16 (6.7) | 0.873 |
| **Apgar score <7 at 5 minutes** | 6 (1.2) | 3 (1.3) | 1.0* |
| **Umbilical artery lactate ≥ 6.0 mmol/l:**<br>291 (58.3); 108 (45.2) | 62 (21.3) | 28 (25.5) | 0.327 |
| **Birth Weight:** | | | 0.170 |
| <3000g | 141 (28.3) | 60 (25.1) | |
| 3000-4000g | 306 (61.3) | 162 (67.8) | |
| >4000g | 52 (10.4) | 17 (7.1) | |
| **Customised growth centile >90th** | 31 (6.2) | 10 (4.2) | 0.260 |
| **Head circumference (cm) mean ±SD:**<br>481 (96); 231 (96.7) | 34.7 ±SD 1.54 | 34.7 ± SD 1.57 | 0.991 |
| **Length (cm) mean ±SD:**<br>482 (96.6); 232 (97.1) | 51.2 ± SD 2.56 | 51.3 ± SD 2.43 | 0.624 |
| **Neonatal Intensive Care Unit admission** | 25 (5.0) | 10 (4.2) | 0.133 |
| **Respiratory distress syndrome ‡** | 5 (1.0) | 4 (1.7) | 0.481* |
| **Non-specific respiratory distress** | 6 (1.2) | 2 (0.8) | 1.0* |
| **Cranial haemorrhage §** | 8 (1.6) | 9 (3.8) | 0.067 |
| **Hypoglycaemia (<2.6mmol/l)** | 33 (6.6) | 13 (5.4) | 0.537 |
| **Jaundice requiring phototherapy** | 20 (4.0) | 4 (1.7) | 0.120* |
| **NICU stay (days):**<br>25 (5.0); 10 (4.2) | | | 0.661* |
| ≤ 7 days | 20 (80.0) | 7 (70.0) | |
| >7 days | 5 (20.0) | 3 (30.0) | |
| **Feeding on discharge:**<br>303 (60.7); 126 (52.7) | | | 0.520* |
| Full/Exclusive breastfeeding | 209 (69.0) | 93 (73.8) | |
| Partially breastfeeding | 88 (29.0) | 30 (23.8) | |
| Artificial formula | 7 (2.3) | 2 (1.6) | |

†Total OP/OT n (%), OA n (%) for variable where it differs from whole sample.

‡ANZNN classification.

§Includes intracranial and extracranial haemorrhage.

* Fisher Exact P value for small values.

sometimes be difficult to read on normal settings, they became readable by magnifying the script.

Maternal age, parity, and previous caesarean were not associated with malposition, consistent with findings from other studies investigating malposition in early labour [2, 28, 30], as opposed to studies assessing fetal malposition during second stage labour which did report an association with these factors [2, 8, 12, 14].

Increased BMI was the only maternal factor found to be associated with fetal malposition. The association of higher maternal BMI with fetal malposition is consistent with existing research [2, 12], including a large retrospective study of 30,839 women with persistent OP [12]. Other studies have found no relationship between BMI and fetal malposition [15, 28]. The relationship between a high BMI and fetal malposition is unclear. Sedentary postures, sometimes favoured by women who are obese or over weight, are associated with posterior

fetal rotation [17] and reclining postures may encourage malposition through the application of gravitational forces on the fetal spine [31].

The malpositioned fetal occiput was more commonly located on the right side in this study, consistent with previous studies [4, 5]. An anterior placenta was not associated with OP/OT in labour in this study, contrary to the findings of other studies [11, 13, 28]. However, placental position information was missing for 17% of women, reflecting those receiving private scans in the community, which may have influenced this finding. The convex structure and location of anterior placentae has been suggested to impede anterior rotation of the fetal spine [13].

Induction of labour was not associated with malposition, consistent with the findings of a previous study [12]. However, in contrast, this study saw an increased use of oxytocin to augment labour after excluding women who had an induction of labour, possibly related to reports of improved anterior rotation with the use of oxytocin [5]. Other evidence related to the use of IOL and oxytocin augmentation with fetal malposition is inconsistent [2, 8, 11, 14, 15]. Similar to previous research [12], artificial ROM was more prevalent after controlling for oxytocin use yet differed in this study as artificial ROM was not significant after controlling for IOL.

This study's finding that malposition resulted in a longer first stage labour has previously been reported [12, 14]. The study saw no association with a longer second stage (median hours: minutes 1:10, OP/OT group; 1:05, OA group) as previously reported [30], contrary to other reports [14–16]. This likely reflects the 74% rate of anterior rotation in the OP/OT group and that 80% of those with a persistent malposition at birth had a caesarean section, with significantly more caesareans performed in the first stage of labour. The association of epidural use with fetal malposition is well established [2, 8, 11, 12, 14] and likely relates to the longer length of labour reported here and by other studies [8, 12, 14] and maternal back pain [4, 13].

The rate of anterior rotation (74%) by the time of birth is similar to other studies [2, 3]. Consistent with published research, our study found birth following a malposition in labour was more likely by caesarean section [2, 3, 8, 11, 13–15], and less likely by normal vaginal birth [8, 14, 15]. Fetal malposition was associated with PPH secondary to caesarean section, similar to another study [32], although not following vaginal birth [12]. Overall perineal trauma was not associated with fetal malposition as found in a recent study [15], although other studies have reported a high prevalence of perineal trauma associated with malposition [8, 12, 14]. It is not clear why there were fewer postnatal urinary catheterisations in the OP/OT group.

Neonatal outcomes were similar in women with a fetal malposition compared to women with no malposition in this study, consistent with some previous research [8, 15]. This is in contrast to earlier large cohort studies that reported adverse neonatal outcomes associated with persistent fetal malposition in the second stage of labour including NICU admission [18, 33], birth injury [20], Apgar score <7 at 5 minutes [33], and low cord pH [18]. The rate of anterior rotation during labour in the OP/OT group may explain the similar neonatal outcomes between groups in this study. Birthweight was not associated with malposition in labour, consistent with several other studies [8, 13, 14].

Differences in outcomes of malposition in the first stage of labour compared to the second stage could be the subject of further research. In addition, further research exploring effective maternal posture interventions and the impact of gravity on the fetal spine to correct sonographically confirmed fetal malposition during labour [34] could lead to significant improvements in maternal health.

## Acknowledgments

We thank Professor Christopher Triggs of the Department of Statistics, University of Auckland, for statistical support.

## Author Contributions

**Conceptualization:** Jennifer Barrowclough.

**Data curation:** Jennifer Barrowclough.

**Formal analysis:** Jennifer Barrowclough.

**Investigation:** Jennifer Barrowclough.

**Methodology:** Jennifer Barrowclough, Bridget Kool, Caroline Crowther.

**Project administration:** Jennifer Barrowclough.

**Supervision:** Bridget Kool, Caroline Crowther.

**Validation:** Bridget Kool, Caroline Crowther.

**Writing – original draft:** Jennifer Barrowclough.

**Writing – review & editing:** Jennifer Barrowclough, Bridget Kool, Caroline Crowther.

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
