## [Decision Letter · Decision Letter 0]

26 Jul 2022

PONE-D-22-12486Fetal malposition in labour and health outcomes for women and their infants: A retrospective cohort studyPLOS ONE

Dear Dr. Barrowclough,

Thank you for submitting your manuscript to PLOS ONE. After careful consideration, we feel that it has merit but does not fully meet PLOS ONE’s publication criteria as it currently stands. Therefore, we invite you to submit a revised version of the manuscript that addresses the points raised during the review process.

We look forward to receiving your revised manuscript.

Kind regards,

Nnabuike Chibuoke Ngene, Dip HIV Med; MMed(FamMed); FCOG; MMed(O&G); Ph.D

Academic Editor

PLOS ONE

Journal Requirements:

2. You indicated that you had ethical approval for your study. In your Methods section, please ensure you have also stated whether you obtained consent from parents or guardians of the minors included in the study or whether the research ethics committee or IRB specifically waived the need for their consent

Additional Editor Comments:

This is a retrospective study that assessed the maternal and neonatal outcomes of fetal position during labour. The study showed that 449/738 (67.6%) of the foetuses were in OP/OT position during the first stage of labour. The foetuses in OP/OT position at birth were 127 and 9 of foetuses previously in OP/OT and OA positions respectively (Table 3). It is an important topic. The authors should respond to the following comments.

1. Abstract, results: “Fetal malposition appears to be more likely in women with a right-sided fetus.” Is right-sided fetus describing a fetus whose body occupies the right side of the uterus?

2. Abstract, results: “Fetal malposition during labour was not associated with adverse infant outcomes.” For how long were the babies followed-up? Infant outcomes suggest that the baby was followed up for one year following childbirth. Use an adjective that will best describe the follow-up period and make necessary changes in the title and other sections of the manuscript. For instance, the word infants in the title may be replaced with neonates.

3. Abstract, conclusion: Explain the intervention that the women received and how they improved their health. Is this conclusion based on the methods/results of this study?

4. Introduction, first sentence: “Fetal malposition refers to an infant in an occiput-posterior (OP) or occiput-transverse (OT) position in labour.” Is a fetus also called an infant? Use the correct terminology in the manuscript.

5. Introduction: “ Factors associated with fetal malposition include nulliparity [8], an anterior placenta [9], pelvic shape [10], epidural use [11], increased body mass index [2], advanced maternal age and fetal macrosomia [12].” Explain the possible mechanisms through which nulliparity and fetal macrosomia are associated with fetal malposition.

6. Introduction, last sentence: “Therefore, the aim of this study was to assess the incidence and health outcomes for women and their infants with a fetal malposition in the first or second stage of labour.” Again, the word infancy suggests that the babies were followed-up/observed for one year following childbirth. Use an adjective that will best describe the follow-up period.

7. Methods and materials: Data of women aged 16 years and above were included. Explain the ethics about using data of children for research purposes (without consent) in the study setting.

8. Methods and materials: What is “lead maternity carer during pregnancy?” This explanation is required regardless of the content of Table 1.

9. Methods and materials: Provide additional details about “postpartum urinary catheterisation” Include the indications and the reason for making it a variable.

10. Methods and materials: What is occiput left/right location? Is fetal location referring to fetal position?

11. Methods and materials: Regardless of the citation number 24, provide additional details about the area deprivation scale. This will assist readers to understand Table 1.

12. Methods and materials: One of the variables assessed was “hypoglycaemia <2.6 mmol/l prior to discharge” Is it hospital discharge or discharge from the labour ward to postnatal ward?

13. Methods and materials: “cord lactate ≥ 6” was assessed. From what site and vessel were the sample collected from? What is the reason for using a threshold of ≥6? Provide a reference to support the choice of this threshold. Explain the reason for not using fetal umbilical artery cord pH <7.0, and or base deficit ≥12 mmol/L.

14. Methods and materials: Provide additional details about sample size calculation. What formular/software was used for the sample size calculation? What were the input parameters?

15. It is written in the manuscript that 1000 was the sample size (as documented in the section on methods and materials). In the abstract, the authors wrote that only 738 maternity records were randomly selected (see abstract). Provide details about the random selection. If necessary include a flow diagram to make it easier to understand. Clarify if there were exclusion criteria.

16. Clarify the reason for reporting relative risk in this retrospective study. Justify your answer with appropriate reference/s.

17. What level of p-value was used to denote statistical significant difference between groups?

18. Table 1: When was the body mass index measured? Was it during the first prenatal clinic visit, or during labour?

19. Results, first and second sentences: “A total of 1000 women were selected using random sampling, conducted by an independent statistician using R statistical packages (R Core Team 2013), from the 4376 eligible women who gave birth at Hospital during 2018. 262 women were excluded due to no record of fetal position in labour, resulting in a sample of 738 women.” This description should be included in the methods and materials and not result section.

20. In Table 1, what are the reasons for choosing the age categories presented?

21. What are the reasons for choosing the categories of parity reported in the Table 1?

22. In Table 1 “Hospital Team” is one of the categories under “Type of lead maternity carer in pregnancy.” Explain what is hospital Team.

23. In Table 1, under ethnicity, the frequency of American/African was not reported.

24. In Table 1, “499 (100); 237 (99.2)” and “496 (99.4); 237 (99.2)” were written under Area deprivation scale and BMI respectively. What do they represent? Provide similar explanation in Tables 2 – 4 where applicable.

25. In Table 1, some of the frequencies do not add up, with some being less than 99.5%. For example, BMI under OA in labour. Check the correctness of your sum/approximation in Tables 1 – 4.

26. Calculate the p-value of each variable in Table 1. This will show if the characteristics of the women in each column are the same or different.

27. In the results, Labour characteristics, the authors wrote: Women in the OP/OT group… have an epidural for analgesia (74% cf. 61%, P<0.001)) than women in the OA group (Table 2).” One of the closing brackets is double.

28. State the meaning of RR and MD in Table 2?

29. In the results, Labour characteristics, sentence: “After controlling for confounding by oxytocin augmentation, artificial ROM was associated with fetal malposition (P=0.050).” In the methods and materials (in the paragraph that deals with statistical analysis), name the confounders and provide step by step details of the statistics applied to controlled them.

30. What is the importance of “Total n(%) OP/OT; OA” in the heading of the first column in the various tables?

31. In Table 3, the frequency of “caesarean indication” is more than the number of “emergency caesarean section.” Ensure that the frequencies add-up.

32. Discussion, first sentence: “The overall incidence of malposition was 68% in labour and 18% at birth.” In the results section, explain how 18% was calculated/observed. This is because Table 3 showed that 449/738 (67.6%) of the foetuses were in OP/OT position during the first stage of labour. The same Table 3 showed that the foetuses in OP/OT position at birth were 127 and 9 of foetuses previously in OP/OT and OA positions respectively.

33. Explain the possible reasons for the greater frequency of occurrence of OP/OT than OA position during labour in the index study.

34. Discussion, second paragraph, sentence: “A strength of this study is the novel approach of assessing outcomes of OP/OT occurring during first and second stage labour rather than only those with a persistent OP/OT in second stage labour or at birth, allowing assessment of outcomes before a spontaneous anterior rotation.” Revise this sentence because other authors have previously studied and reported on the outcomes of OP position during the first and second stages of labour. For example, Martino V, Iliceto N, Simeoni U. Occipito-posterior fetal head position, maternal and neonatal outcome. Minerva Ginecol. 2007 Aug;59(4):459-64. PMID: 17923836. https://pubmed.ncbi.nlm.nih.gov/17923836/.

35. Discussion, third paragraph, sentence: “Data were sometimes not available for some of the study outcomes due to an absence of documentation in the medical records.” Explain the reason for not reporting missing data.

36. Discussion, seventh paragraph, sentence: “However, in contrast, this study saw an increased use of oxytocin to augment labour after excluding IOL,…” “Excluding IOL” is difficult to understand. Revise the sentence.

37. What is the explanation for the fewer postnatal catheterisations that were performed for women in the OP/OT group (9.6% cf.14.6% OA, P=0.043).

38. Are there measures to prevent OP position.

Reviewers' comments:

Reviewer's Responses to Questions

**Comments to the Author**

1. Is the manuscript technically sound, and do the data support the conclusions?

Reviewer #1: Partly

Reviewer #2: Yes

2. Has the statistical analysis been performed appropriately and rigorously? 

Reviewer #1: Yes

Reviewer #2: Yes

3. Have the authors made all data underlying the findings in their manuscript fully available?

Reviewer #1: Yes

Reviewer #2: Yes

4. Is the manuscript presented in an intelligible fashion and written in standard English?

Reviewer #1: Yes

Reviewer #2: Yes

5. Review Comments to the Author

Reviewer #1: The authors have written a retrospective review comparing fetal occiput positions and subsequent outcomes during the first and second stages of labor.

Abstract:

Succinct and understandable. The conclusion sentence does not match the aims sentence and should be re-written.

Introduction:

The authors give background information that argues for the further study of 1st stage of labor malposition and that prior studies have focused on persistent malposition in the second stage. However, in this study the population examined is either in the 1st or second stage of labor. Why did the authors decide to include 2nd stage malposition?

Method and Materials:

line 57- consider changing the word "jiggling" to another word.

I think there needs to be more added about when exams were done, by whom and how they were verified. This study hinges on the qualitative digital exam of the fetal occiput. Who did the exam, at what part of the first stage of labor (or anytime therein?). were any exams verified by ultrasound. Were exams done by experienced providers or trainees? I know the authors include women at> 3 cm, but were most 1st stage of labor exams at 9cm? or were more at 3? Were the 1st stage of labor exams done in early or active labor? If these authors dont have this information, I believe their discussion should include lack of this information as a limitation.

Did the authors consider comparing 1st stage malposition versus persistent second stage malposition with regards to perinatal outcomes?

Were adverse outcomes different between fetuses who were malpositioned in the 1st stage of labor, but then did not persist in the 2nd stage?

Results:

It is surprising that the length of the second stage of labor did not differ based on malposition. Why do the authors think that is? What was the median duration of second stage in both groups?

What were the indications for operative delivery?

Discussion:

The authors not the inaccuracy of digital examination for fetal position. They should add a sentence about how ultrasound confirmation is superior. Perhaps a future study can look at ultrasound verified fetal malposition.

Reviewer #2: This is an interesting an well executed study. Since it is a retrospective study where the data was gathered from the clinical documents, how complete and accurate was the information? in clinical practice, patients are managed by midwives, medical students and doctors of different level of training and experience. Furthermore, handwritings differ and sometimes it might be impossible to read what was written. Under the limitations of this study the authors must add the problems experienced in this respect.

6. PLOS authors have the option to publish the peer review history of their article (what does this mean?). If published, this will include your full peer review and any attached files.

Reviewer #1: No

Reviewer #2: No

---

## [Author Response · Author response to Decision Letter 0]

25 Sep 2022

Response to Reviewers

Editors comments:

1. Abstract, results: “Fetal malposition appears to be more likely in women with a right-sided fetus.” Is right-sided fetus describing a fetus whose body occupies the right side of the uterus?

Response: ‘Right-sided fetus’ has been replaced with ‘right-sided fetal occiput’ (line 16).

2. Abstract, results: “Fetal malposition during labour was not associated with adverse infant outcomes.” For how long were the babies followed-up? Infant outcomes suggest that the baby was followed up for one year following childbirth. Use an adjective that will best describe the follow-up period and make necessary changes in the title and other sections of the manuscript. For instance, the word infants in the title may be replaced with neonates.

Response: The babies were followed up until hospital discharge. The word infant has been replaced with ‘neonate’ in all relevant places and ‘newborn infant’ wherever it previously referred to women and their infant.

3. Abstract, conclusion: Explain the intervention that the women received and how they improved their health. 

Response: This was an observational study so no specific intervention was applied other than their usual care. 

Is this conclusion based on the methods/results of this study?

Response: Yes, because maternal outcomes in the first and second stage of labour were associated with malposition, they could be improved if effective labour interventions are applied, as opposed to waiting till birth to intervene. 

4. Introduction, first sentence: “Fetal malposition refers to an infant in an occiput-posterior (OP) or occiput-transverse (OT) position in labour.” Is a fetus also called an infant? Use the correct terminology in the manuscript. 

Response: ‘infant’ has been replaced with ‘fetus’ in line 24 as fetus was intended.

5. Introduction: “ Factors associated with fetal malposition include nulliparity [8], an anterior placenta [9], pelvic shape [10], epidural use [11], increased body mass index [2], advanced maternal age and fetal macrosomia [12].” Explain the possible mechanisms through which

nulliparity and fetal macrosomia are associated with fetal malposition. 

Response: The text ‘Restricted space for anterior fetal rotation may occur due to strong abdominal muscles in nulliparous women or fetal macrosomia. Alternatively, slower progression to full cervical dilatation and therefore exposure of the fetal head to counterpressure of the pelvic floor is another possible mechanism by which nulliparity is associated with persistent malposition.’ has been added to lines 31 – 34.

6. Introduction, last sentence: “Therefore, the aim of this study was to assess the incidence and health outcomes for women and their infants with a fetal malposition in the first or second stage of labour.” Again, the word infancy suggests that the babies were followed up/observed for one year following childbirth. Use an adjective that will best describe the follow-up period. 

Response: ‘infants’ has been preceded with the word ‘newborn’ in line 47.

7. Methods and materials: Data of women aged 16 years and above were included. Explain the ethics about using data of children for research purposes (without consent) in the study setting. 

Response: The following text has been added ‘Ethics approval was provided for the use of data from women aged 16 years+ without consent because the sharing of that data is covered by Auckland District Health Board’s data access policy. The data was de-identified during analysis and reporting, and is only used for research purposes intended to benefit maternal and neonatal health in accordance with the Declaration of Helsinki 1996.’ (lines 106 – 110).

8. Methods and materials: What is “lead maternity carer during pregnancy?” This explanation is required regardless of the content of Table 1. 

Response: The phrase ‘midwife or obstetric specialist contracted to provide maternity care’ has been added in parentheses (line 61).

9. Methods and materials: Provide additional details about “postpartum urinary catheterisation” Include the indications and the reason for making it a variable. 

Response: ‘any’ now precedes ‘postpartum’ (line 69) and the phrase ‘as a measure of maternal complications’ has need added in parentheses (line 70).

10. Methods and materials: What is occiput left/right location? Is fetal location referring to fetal position? 

Response: This has been re-worded and now reads ‘left/right-sided fetal occipital location’ (lines 64 - 65).

11. Methods and materials: Regardless of the citation number 24, provide additional details about the area deprivation scale. This will assist readers to understand Table 1. 

Response: The following sentence has been added ‘The NZDep is an area-based measure of socioeconomic deprivation in New Zealand. Decile 1 represents areas with the least deprivation, Decile 10 represents areas with the most deprivation.’ (lines 79 - 81).

12. Methods and materials: One of the variables assessed was “hypoglycaemia <2.6 mmol/l prior to discharge” Is it hospital discharge or discharge from the labour ward to postnatal ward? 

Response: The period that the hypoglycaemia refers to is until hospital discharge, this has now been clarified in line 75.

13. Methods and materials: “cord lactate ≥ 6” was assessed. From what site and vessel were the sample collected from? What is the reason for using a threshold of ≥6? Provide a reference to support the choice of this threshold. Explain the reason for not using fetal umbilical artery cord pH <7.0, and or base deficit ≥12 mmol/L. 

Response: The sample for cord lactate levels were collected from umbilical arterial vessels. The phrase ‘cord lactate’ has been replaced with ‘umbilical arterial lactate’ in line 73. In addition, the revised wording has been applied to the Results section (lines 202 - 203) and Table 4. The sentence ‘Abnormal arterial cord lactates, considered a more accurate measure of metabolic acidosis than umbilical artery pH <7.0 and Base Excess <-12 mmol/l alone, were the standard measure of fetal acidosis at the hospital in 2018 with a clinical cutoff of ≥ 6 mmol/l, though revised to ≥ 6.1 mmol/l in 2019 [26].’ has been added including the reference, in lines 77 - 79.

14. Methods and materials: Provide additional details about sample size calculation. What formular/software was used for the sample size calculation? What were the input parameters? 

Response: ‘ClinCalc.com’ was the software used for sample size calculations, this detail has been added to line 84. The input parameters are provided in line 85 i.e. a 16% incidence of caesarean section in the OP/OT group and a 10% incidence in the OA group with an accompanying citation for Akmal et al 2004 BJOG. 

15. It is written in the manuscript that 1000 was the sample size (as documented in the section on methods and materials). In the abstract, the authors wrote that only 738 maternity records were randomly selected (see abstract). Provide details about the random selection. If necessary include a flow diagram to make it easier to understand. Clarify if there were exclusion criteria. 

Response: Details about the random selection of records have been moved from the Results to the Methods and Materials of the main text. Exclusion criteria are described in lines 56-57 of the Methods and Materials i.e. ‘Women with a major fetal abnormality were not included’. 

16. Clarify the reason for reporting relative risk in this retrospective study. Justify your answer with appropriate reference/s. 

Response: We chose to use risk ratio as the comparative statistic in the cohort study as it is considered appropriate (Grimes, D. A., & Schulz, K. F. (2008). Making Sense of Odds and Odds Ratios. 111(2 Part 1), 423-426. 10.1097/01.AOG.0000297304.32187.5d; Cummings P. The Relative Merits of Risk Ratios and Odds Ratios Arch Pediatr Adolesc Med. 2009;163(5):438-445. doi:10.1001/archpediatrics.2009.31)

17. What level of p-value was used to denote statistical significant difference between groups? 

Response: ‘A P-value ≤ 0.05 was used to denote a statistically significant difference.’ has been added (line 98).

18. Table 1: When was the body mass index measured? Was it during the first prenatal clinic visit, or during labour? 

Response: BMI was measured at the initial prenatal clinic visit, this has now been clarified in the Methods and Materials (line 60).

19. Results, first and second sentences: “A total of 1000 women were selected using random sampling, conducted by an independent statistician using R statistical packages (R Core Team 2013), from the 4376 eligible women who gave birth at Hospital during 2018. 262 women were excluded due to no record of fetal position in labour, resulting in a sample of 738 women.” This description should be included in the methods and materials and not result section. 

Response: These two sentences have been moved to the Methods and Materials (lines 86 – 90). The second sentence has been revised to ‘However, 262 maternity records had no record of fetal position in labour so were unable to provide data, resulting in a sample of 738 women.’

20. In Table 1, what are the reasons for choosing the age categories presented? 

Response: Use of these categories reflects the distribution of maternal age in the sample.

21. What are the reasons for choosing the categories of parity reported in the Table 1? 

Response: The categories for parity were chosen as nulliparity is evidenced as a risk factor for malposition, and those with a parity of 3 or more may be at risk of unstable fetal position due to stretched musculature.

22. In Table 1 “Hospital Team” is one of the categories under “Type of lead maternity carer in pregnancy.” Explain what is hospital Team. 

Response: The phrase ‘Midwifery/obstetric team supervised by a senior medical officer’ have been added to Table 1 footnotes following the symbol ¶ which is also added to the variable in the table (line 129).

23. In Table 1, under ethnicity, the frequency of American/African was not reported. 

Response: American/African is part of the MELAA category (Middle Eastern, Latin American, African) but was spread across two lines. The table has been revised so the full variable name is now displayed on one line.

24. In Table 1, “499 (100); 237 (99.2)” and “496 (99.4); 237 (99.2)” were written under Area deprivation scale and BMI respectively. What do they represent? Provide similar explanation in Tables 2 – 4 where applicable. 

Response: These represent the total n(%) for the variable wherever this differs from the whole sample. An explanation has been added in the footnotes of each table with the symbol † in the left column header.

25. In Table 1, some of the frequencies do not add up, with some being less than 99.5%. For example, BMI under OA in labour. Check the correctness of your sum/approximation in Tables 1 – 4. 

Response: Thank you for highlighting this error. These frequencies have been checked and BMI percentages corrected.

26. Calculate the p-value of each variable in Table 1. This will show if the characteristics of the women in each column are the same or different. 

Response: The P-values have been added to Table 1.

27. In the results, Labour characteristics, the authors wrote: Women in the OP/OT group...have an epidural for analgesia (74% cf. 61%, P<0.001)) than women in the OA group (Table 2).” One of the closing brackets is double. 

Response: Thank you for highlighting this, the redundant closing bracket has been deleted (line 134).

28. State the meaning of RR and MD in Table 2? 

Response: RR and MD have now been written in full in the column header.

29. In the results, Labour characteristics, sentence: “After controlling for confounding by oxytocin augmentation, artificial ROM was associated with fetal malposition (P=0.050).” In the methods and materials (in the paragraph that deals with statistical analysis), name the

confounders and provide step by step details of the statistics applied to controlled them. 

Response: A sentence has been added to the Methods and Materials (statistical analysis paragraph) naming the confounder oxytocin augmentation, for the variable artificial rupture of membranes, and statistical method to control for it (lines 98 - 102).

30. What is the importance of “Total n(%) OP/OT; OA” in the heading of the first column in the various tables? This was to indicate those variables that had a different sample size to column 2 & 3. 

Response: For clarity and to address your point 24 above, the tables now have a symbol after heading 1 in the first column with the footnote ‘† Total OP/OT n (%), OA n (%) for variable where different from whole sample’.

31. In Table 3, the frequency of “caesarean indication” is more than the number of “emergency caesarean section.” Ensure that the frequencies add-up. 

Response: We have checked this and the ‘Mode of birth: Emergency caesarean section' numbers (119 and 19) mirror the numbers in the 'Caesarean indication’ (119 and 19).

32. Discussion, first sentence: “The overall incidence of malposition was 68% in labour and 18% at birth.” In the results section, explain how 18% was calculated/observed. This is because Table 3 showed that 449/738 (67.6%) of the foetuses were in OP/OT position during

the first stage of labour. The same Table 3 showed that the foetuses in OP/OT position at birth were 127 and 9 of foetuses previously in OP/OT and OA positions respectively. 

Response: The following sentence has been added to the Results section ‘Overall, 18% of fetuses (n=136/738) from both the OP/OT group (n=127) and OA group (n=9) were in an OP/OT malposition at birth.’ (lines 166 – 167) 

33. Explain the possible reasons for the greater frequency of occurrence of OP/OT than OA position during labour in the index study. 

Response: The following text has been added ‘Possible reasons for the greater frequency of OP/OT position compared to OA position during labour include the presence of risk factors associated with fetal malposition, and the presence of malposition at the establishment of induced labour [28], a process that often involves 24 - 48 hours of inpatient care for fetal monitoring commonly in semi-recumbent postures in which gravity may encourage posterior rotation of the fetal spine.’ (lines 213 - 218).

34. Discussion, second paragraph, sentence: “A strength of this study is the novel approach of assessing outcomes of OP/OT occurring during first and second stage labour rather than only those with a persistent OP/OT in second stage labour or at birth, allowing assessment of outcomes before a spontaneous anterior rotation.” Revise this sentence because other authors have previously studied and reported on the outcomes of OP position during the first and second stages of labour. For example, Martino V, Iliceto N, Simeoni U. Occipito-posterior

fetal head position, maternal and neonatal outcome. Minerva Ginecol. 2007 Aug;59(4):459-64. PMID: 17923836. https://pubmed.ncbi.nlm.nih.gov/17923836/.

 Response: Thank you, we do understand why the wording might be confusing. While studies discussed in the paper by Martino et al have reported on outcomes during the first and second stage of labour, the fetal malposition was determined at birth or in the second stage of labour, and therefore may not reflect women who had a fetal malposition in the first stage before anterior fetal rotation. The studies by Neri et al (1995) and Gardberg et al (1994) report on outcomes of persistent malposition that was diagnosed in the second stage of labour. There is a paucity of evidence which dates back 20 years reporting outcomes related to malposition during the first stage as mentioned in the background (lines 42 – 44) referring to duration of labour and caesarean section. In light of your feedback, the sentence has been rephrased as ‘A strength of this study is the novel approach of assessing outcomes related to malposition that is diagnosed during first and second stage labour rather than only a persistent OP/OT in the second stage labour or at birth,’ (lines 225 – 227).

35. Discussion, third paragraph, sentence: “Data were sometimes not available for some of the study outcomes due to an absence of documentation in the medical records.” Explain the reason for not reporting missing data. 

Response: To provide further clarity, the following sentence has been added ‘While missing data is not reported as a subset category, the denominators reflect the totals minus the missing data.’ (lines 243-245).

36. Discussion, seventh paragraph, sentence: “However, in contrast, this study saw an increased use of oxytocin to augment labour after excluding IOL,…” “Excluding IOL” is difficult to understand. Revise the sentence. 

Response: The sentence has been revised to “…after excluding women who had an induction of labour.’ (line 265).

37. What is the explanation for the fewer postnatal catheterisations that were performed for women in the OP/OT group (9.6% cf.14.6% OA, P=0.043). 

Response: The following sentence has been added ‘It is not clear why there were fewer postnatal urinary catheterisations in the OP/OT group.’ (lines 284 – 285).

38. Are there measures to prevent OP position. 

Response: The last sentence has been revised to ‘In addition, further research exploring effective maternal posture interventions and the impact of gravity on the fetal spine to correct sonographically confirmed fetal malposition during labour [34] could lead to significant improvements in maternal health.’ (lines 295 – 298).

Reviewer #1

Abstract:

The conclusion sentence does not match the aims sentence and should be re-written. 

Response: We have revised the Conclusion to read ‘Interventions such as maternal posture in the first and second stage of labour could potentially reduce the incidence of malposition and improve health outcomes for mothers.’ (lines 20-22).

Introduction:

Why did the authors decide to include 2nd stage malposition? 

Response: We considered it important to have evidence of the incidences and outcomes of malposition at any time in labour so that the full extent of malposition on a continuum is reflected in a single sample, given that an occipito-anterior fetus may become an occipito-posterior fetus in the second stage in women with an epidural (Lieberman 2005) and that studies of malposition in the second stage do not reflect labour outcomes such as abnormal fetal heart rate in women who have a caesarean section in the first stage. Therefore, a single study that includes malposition at any stage in labour provides a comprehensive assessment of outcomes of malposition in labour which is comprehensible and useful to trialists, healthcare providers and consumers. 

Methods and materials:

Line 57 – consider changing the word ‘jiggling’ to another word.

Response: 'Jiggling' is a commonly used descriptor associated with the use of the Rebozo technique and therefore the author’s preference is to retain it (line 65). Reference: https://onlinelibrary.wiley.com/doi/full/10.1111/jmwh.12352

I think there needs to be more added about when exams were done, by whom and how they

were verified. This study hinges on the qualitative digital exam of the fetal occiput. Who did

the exam, at what part of the first stage of labor (or anytime therein?). were any exams verified

by ultrasound. Were exams done by experienced providers or trainees? I know the authors

include women at> 3 cm, but were most 1st stage of labor exams at 9cm? or were more at 3?

Were the 1st stage of labor exams done in early or active labor? If these authors dont have this

information, I believe their discussion should include lack of this information as a limitation.

Response: Midwives or obstetricians performed the vaginal exams. Women were in established labour. Most vaginal exams were performed from 6 cm dilatation but could have been at lower dilatations if labour was established. The Methods and Materials state the method of diagnosis was by vaginal examination. Due to the retrospective nature of this study, confirmation by ultrasound was not possible. The word ‘established’ now precedes the word “labour” (line 53). The words ‘by a midwife or obstetrician’ now follows the words “vaginal examination” (lines 54-55).

Did the authors consider comparing 1st stage malposition versus persistent second stage

malposition with regards to perinatal outcomes? 

Response: A comparison of first stage malposition and persistent second stage malposition was outside the scope of this study which was to compare outcomes for women with OP/OT position to women with OA position. However, the sentence ‘Differences in outcomes of malposition in the first stage of labour compared to the second stage could be the subject of further research.’ has been added to the section where areas for future research are discussed (lines 294-295).

Were adverse outcomes different between fetuses who were malpositioned in the 1st stage of

labor, but then did not persist in the 2nd stage? 

Response: This was not within the scope of this study, which was a comparative study of OP/OT and OA fetal positions in a single cohort of women from first stage of labour. 

Results:

It is surprising that the length of the second stage of labor did not differ based on malposition.

Why do the authors think that is? 

Response: Thank you for drawing this to our attention, the following sentence has been inserted in the Discussion (lines 273-275) ‘This likely reflects the 74% rate of anterior fetal rotation in the OP/OT group and that 80% of those with a persistent malposition at birth had a caesarean section, with significantly more caesareans performed in the first stage of labour.’ 

What was the median duration of second stage in both groups?

Response: The median duration of the second stage was 1:10 median hours and minutes OP/OT and 1:05 median hours and minutes for OA. We have added this information in lines 271-272.

What were the indications for operative delivery? 

Response: Indications for operative delivery are presented in Table 3 and discussed in the Results. There was no significant difference between the groups.

Discussion: 

The authors not the inaccuracy of digital examination for fetal position. They should add a sentence about how ultrasound confirmation is superior. Perhaps a future study can look at

ultrasound verified fetal malposition.

Response: In the Discussion we refer to the inaccuracy of diagnosis by digital examination compared to sonographic diagnosis in lines 233 and 240. The words ‘sonographically confirmed’ now precede “fetal position” in the last sentence of the Discussion (lines 296-297).

Reviewer #2

Since it is a retrospective study where the data was gathered from the clinical documents, how complete and accurate was the information? in clinical practice, patients are managed by midwives, medical students and doctors of different level of training and experience. Furthermore, handwritings differ and sometimes it might be impossible to read what was written. Under the limitations of this study the authors must add the problems experienced in this respect.

Response: Totals for all outcomes reflect the data that was available for each outcome. The following sentence ‘Whilst missing data are not reported as a subset category, the denominators reflect the totals minus the missing data.’ has been added, according to the editors comment, to lines 243-244. Missing data for location of placenta was fairly common due to the number of women under private care who often have their ultrasound scans in the community and the data is not then shared with the hospital. The text ‘, reflecting those receiving private scans in the community,’ has been inserted into the sentence discussing this in the Discussion (lines 260-261). All entries of vaginal examinations in the maternity records at Auckland Hospital are verified by a qualified midwife or performed by obstetric registrars. The Methods and Materials now have the words ‘by a midwife or obstetrician’ follow the words “vaginal examination” in lines 54-55, as per Reviewer #1’s comment. The following sentence has been added to the limitations, ‘Though hand-written notes could sometimes be difficult to read on normal settings, they became readable by magnifying the script.’ (lines 244-245).

---

## [Decision Letter · Decision Letter 1]

6 Oct 2022

Fetal malposition in labour and health outcomes for women and their newborn infants: A retrospective cohort study

PONE-D-22-12486R1

Dear Dr. Barrowclough,

We’re pleased to inform you that your manuscript has been judged scientifically suitable for publication and will be formally accepted for publication once it meets all outstanding technical requirements.

Kind regards,

Nnabuike Chibuoke Ngene, Dip HIV Med; MMed(FamMed); FCOG; MMed(O&G); Ph.D

Academic Editor

PLOS ONE

Additional Editor Comments (optional):

Reviewers' comments:

Reviewer's Responses to Questions

**Comments to the Author**

1. If the authors have adequately addressed your comments raised in a previous round of review and you feel that this manuscript is now acceptable for publication, you may indicate that here to bypass the “Comments to the Author” section, enter your conflict of interest statement in the “Confidential to Editor” section, and submit your "Accept" recommendation.

Reviewer #1: All comments have been addressed

Reviewer #2: All comments have been addressed

2. Is the manuscript technically sound, and do the data support the conclusions?

Reviewer #1: Yes

Reviewer #2: Yes

3. Has the statistical analysis been performed appropriately and rigorously? 

Reviewer #1: Yes

Reviewer #2: N/A

4. Have the authors made all data underlying the findings in their manuscript fully available?

Reviewer #1: Yes

Reviewer #2: Yes

5. Is the manuscript presented in an intelligible fashion and written in standard English?

Reviewer #1: Yes

Reviewer #2: Yes

6. Review Comments to the Author

Reviewer #1: (No Response)

Reviewer #2: No further comments. The authors addended to all comments by the reviewers. I don't think further recommendations are necessary.

7. PLOS authors have the option to publish the peer review history of their article (what does this mean?). If published, this will include your full peer review and any attached files.

Reviewer #1: No

Reviewer #2: No

---

## [Editor Report · Acceptance letter]

10 Oct 2022

PONE-D-22-12486R1 

Fetal malposition in labour and health outcomes for women and their newborn infants: A retrospective cohort study 

Dear Dr. Barrowclough:

I'm pleased to inform you that your manuscript has been deemed suitable for publication in PLOS ONE. Congratulations! Your manuscript is now with our production department. 

Kind regards, 

on behalf of

Dr. Nnabuike Chibuoke Ngene 

Academic Editor

PLOS ONE